# Canonical PRC1 controls sequence-independent propagation of Polycomb-mediated gene silencing

Hagar F. Moussa[1,7], Daniel Bsteh[1,2,7], Ramesh Yelagandula[1], Carina Pribitzer[1], Karin Stecher[1], Katarina Bartalska[1,5], Luca Michetti[1], Jingkui Wang[3], Jorge A. Zepeda-Martinez[1], Ulrich Elling [1], Jacob I. Stuckey[4,6], Lindsey I. James [4], Stephen V. Frye [4] & Oliver Bell [1,2]

Polycomb group (PcG) proteins play critical roles in the epigenetic inheritance of cell fate. The Polycomb Repressive Complexes PRC1 and PRC2 catalyse distinct chromatin modifications to enforce gene silencing, but how transcriptional repression is propagated through mitotic cell divisions remains a key unresolved question. Using reversible tethering of PcG proteins to ectopic sites in mouse embryonic stem cells, here we show that PRC1 can trigger transcriptional repression and Polycomb-dependent chromatin modifications. We find that canonical PRC1 (cPRC1), but not variant PRC1, maintains gene silencing through cell division upon reversal of tethering. Propagation of gene repression is sustained by cis-acting histone modifications, PRC2-mediated H3K27me3 and cPRC1-mediated H2AK119ub1, promoting a sequence-independent feedback mechanism for PcG protein recruitment. Thus, the distinct PRC1 complexes present in vertebrates can differentially regulate epigenetic maintenance of gene silencing, potentially enabling dynamic heritable responses to complex stimuli. Our findings reveal how PcG repression is potentially inherited in vertebrates.

[1] Institute of Molecular Biotechnology of the Austrian Academy of Sciences (IMBA), Vienna Biocenter (VBC), Dr. Bohr-Gasse 3, 1030 Vienna, Austria. [2] Department of Biochemistry and Molecular Medicine and Norris Comprehensive Cancer Center, Keck School of Medicine of the University of Southern California, Los Angeles, CA 90089, USA. [3] Research Institute of Molecular Pathology (IMP), Vienna Biocenter (VBC), Campus-Vienna-Biocenter 1, 1030 Vienna, Austria. [4] Center for Integrative Chemical Biology and Drug Discovery, Division of Chemical Biology and Medicinal Chemistry, UNC Eshelman School of Pharmacy, University of North Carolina at Chapel Hill, Chapel Hill, NC 27599, USA. [5] Present address: Institute of Science and Technology Austria (IST Austria), Am Campus 1, 3400 Klosterneuburg, Austria. [6] Present address: Constellation Pharmaceuticals, 215 First Street, Suite 200, Cambridge, MA 02142, USA. [7] These authors contributed equally: Hagar F. Moussa, Daniel Bsteh. Correspondence and requests for materials should be addressed to O.B. (email: oliver.bell@med.usc.edu)

Epigenetic mechanisms support heritable transmission of differential gene expression patterns, stabilizing diverse cell types in metazoans. Silencing of key developmental genes by Polycomb group (PcG) proteins is arguably the premier paradigm for epigenetic regulation of cell fate inheritance[1]. PcG proteins assemble into distinct multi-subunit complexes with inherent catalytic and non-catalytic activities. Among the two major families, Polycomb Repressive Complex 1 (PRC1) catalyzes monoubiquitination of lysine 119 on histone H2A (H2AK119ub1) and has the capacity to condense nucleosomes[2–4] whereas PRC2 is responsible for mono-, di- and trimethylation of lysine 27 of histone H3(H3K27me2/3)[5]. While these activities are intimately linked to forming and maintaining repressive chromatin domains, the sequence of molecular events underpinning epigenetic repression remain poorly understood[6–8].

In vertebrates, the PRC1 family has diversified into many heterogeneous complexes that can be broadly classified by the presence or absence of Cbx (chromobox-containing protein) subunits. In canonical PRC1 (cPRC1), Cbx confers the ability to bind H3K27me3[9]. This interaction is critical for cPRC1 recruitment to PRC2 target genes and transcriptional silencing[10,11]. Transmission of parental H3K27me3-marked nucleosomes to daughter strands could potentially serve as a *cis*-acting epigenetic signal for propagating Polycomb recruitment and gene repression through DNA replication and cell division[12–14]. However, the role of this histone mark as a carrier of epigenetic information remains controversial as DNA sequence elements contribute to PcG protein targeting in mammals[15–19] and are continuously required for long-term heritable gene silencing in *Drosophila*[20,21]. Variant PRC1 (vPRC1) complexes, which harbor Rybp (Ring1B and Yy1 binding protein), or its homolog Yaf2, are recruited to chromatin independently of H3K27me3[3,22,23]. vPRC1 exclusive targets are only moderately repressed[24], suggesting distinct modes of transcriptional regulation compared to the concerted action of cPRC1 and PRC2. How the different PcG complexes and their chromatin-modifying activities achieve and transmit heritable gene silencing in vertebrates remains unresolved. However, the inability of vPRC1 to interact with H3K27me3 raises the prospect of alternative epigenetic mechanisms.

Here, we use reversible tethering of Cbx7 and Rybp at ectopic loci in mouse embryonic stem cells (mESCs) to compare the capacity of cPRC1 and vPRC1 in formation and inheritance of repressive chromatin domains. Each PcG protein nucleates a distinct functional PRC1 complex that catalyzes repressive chromatin modifications and triggers transcriptional gene silencing. However, after release only Cbx7-initiated repressive chromatin can be maintained through genome replication and cell divisions. We show that sequence-independent propagation of gene silencing requires H3K27me3 and H2AK119ub1 arguing that *cis*-acting histone modifications promote a feedback mechanism for heritable cPRC1 targeting. Together, our findings provide insight into the mechanism of inheritance of Polycomb-dependent repression revealing fundamental differences in the contributions of canonical and vPRC1 complexes.

## Results

**Recruitment of cPRC1 and vPRC1 induces repressive chromatin**. To uncover how cPRC1 and vPRC1 contribute to the initiation and maintenance of repressive chromatin, we engineered mESCs that enable reversible tethering of individual PRC1 complex members to genomic Tet operator (TetO) sites via the Tet$^{OFF}$ system. We generated two mESC lines that express a distinct PRC1 core subunit fused to a FLAG-Tet repressor domain (TetR). TetR DNA binding domain fusions bind to TetO sites, but this sequence-dependent recruitment is abrogated by

addition of Doxycycline (Dox). The cPRC1-TetO line constitutively expresses FLAG-TetR-Cbx7 (TetR-Cbx7), a member of cPRC1. In contrast, the vPRC-TetO line constitutively expresses FLAG-TetR-Rybp (TetR-Rybp), a member of vPRC1 (Fig. 1a and Supplementary Tables 1 and 2). This conditional control of sequence-dependent targeting enables the separation of cause from consequence, and the ability to directly determine the heritable properties of histone modifications in proliferating cells[20,21,25–27]. Based on previous work[28], we hypothesized that the recruitment of core subunits to TetO would facilitate nucleation of functional PRC1 complexes and thus enable a direct comparison of different modifying-activities on the same chromatin template. In addition, because TetR is conditionally released from TetO upon Dox treatment, we could monitor potential differences in heritability of PcG-dependent chromatin modifications and silencing through cell divisions, after loss of the initial stimulus[25–27].

First, we examined if recruitment of different core subunits to an integrated landing site in the genome would direct the assembly of distinct PRC1 complexes and initiate silencing of an adjacent active reporter gene. Western blot analysis of immunoprecipitation using FLAG antibody revealed that TetR-Cbx7 and TetR-Rybp interact with the common catalytic subunit Ring1B. Moreover, TetR-Cbx7, but not TetR-Rybp, interacts with Phc1 suggesting the formation of distinct PRC1 complexes in solution (Supplementary Figs. 1a, b and 2). To evaluate the activity of ectopic PRC1 assemblies, we monitored enrichment of FLAG-tagged proteins, endogenous PcG proteins, and chromatin modifications at a dual reporter gene in TetO-mESCs that contains an array of seven TetO sequences (7× TetO) flanked by a downstream GFP reporter and an upstream BFP reporter (Fig. 1b). Importantly, in the parental line this integration site on chromosome 15 was devoid of active and repressive chromatin marks including PcG-dependent histone modifications (Fig. 1b). Expression of TetR-Cbx7 or -Rybp led to enrichment of Ring1B and H2AK119ub1 not only across 7xTetO but also flanking regions, consistent with spreading of repressive chromatin more than 2 kb in either direction (Fig. 1c and Supplementary Tables 2 and 3). In agreement with results from the co-immunoprecipitation, tethering formed distinct PRC1 complexes, since Rybp was not detected upon tethering of Cbx7 and vice versa. Notably, despite similar Ring1B binding, we observed higher levels of H2AK119ub1 in response to TetR-Rybp recruitment likely reflecting the higher catalytic activity of vPRC1[29]. Further, while tethering of Cbx7 or Rybp conferred distinct assemblies of cPRC1 or vPRC1, respectively, both cases resulted in Suz12 binding and H3K27me3, consistent with a role of H2AK119ub1 in signalling for PRC2 recruitment[28,30,31].

To determine how ectopic recruitment of cPRC1 and vPRC1 affects transcription, we assessed changes in GFP and BFP expression. Flow cytometry indicated that both reporters were highly and homogenously expressed prior to expression of TetR fusions (Fig. 1d and Supplementary Figs. 3a, b and 4a, b). Similarly, expression of TetR DNA binding domain alone had no impact on reporter gene expression (Supplementary Fig. 4c). In contrast, Cbx7- and Rybp-dependent PRC1 assembly was accompanied by complete repression of GFP and BFP consistent with spreading of repressive chromatin modification across both reporter genes (Fig. 1d and Supplementary Fig. 4a).

Thus, ectopic recruitment of core PRC1 subunits is sufficient to nucleate assembly of functionally distinct PRC1 complexes and recapitulate PcG-dependent chromatin modifications that trigger silencing of active transcription. Furthermore, vPRC1- and cPRC1-mediated H2AK119ub1 led to PRC2 recruitment and H3K27me3, establishing a pattern characteristic of endogenous Polycomb target genes and consistent with previous work[28,30,32] (Fig. 1c and Supplementary Fig. 1c).

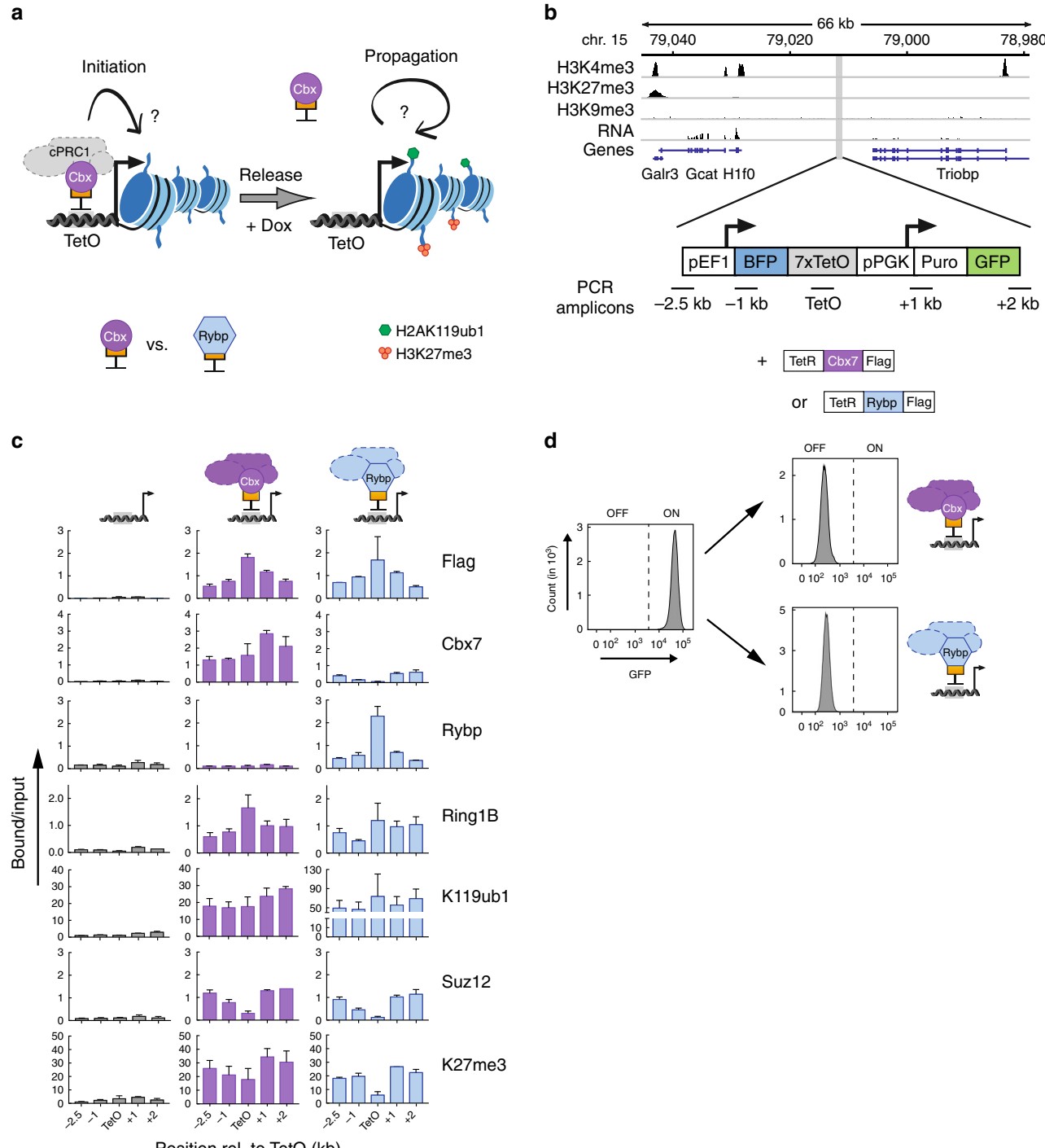

**Fig. 1** cPRC1 and vPRC1 establish repressive chromatin and silence reporter genes. **a** Scheme of experimental design. TetR fusion facilitates reversible tethering of different PcG proteins to Tet Operator sites (TetO) upstream of a reporter gene and tests the consequences of chromatin modifications on transcriptional regulation. Doxycycline (Dox) addition releases TetR binding to determine heritable maintenance of chromatin modifications and expression state in the absence of the initial stimulus. **b** Histone modifications and RNA expression surrounding the integration site of the dual reporter gene construct located on chromosome 15 in mouse ES cells. **c** ChIP-qPCR analysis shows relative enrichments of FLAG-TetR fusion, PcG proteins and histone modifications upstream and downstream of the TetO DNA binding sites (TetO). ChIP enrichments for H2AK119ub1 and H3K27me3 are normalized to negative control locus (IAP). Data are mean ± SD (error bars) of at least two independent experiments. Source data are provided as a Source Data file. **d** Flow cytometry histograms of GFP expression in the absence and presence of TetR PcG fusion proteins

**cPRC1, not vPRC1, promotes sequence-independent inheritance.** Having established that ectopic recruitment of PRC1 subunits is sufficient to initiate gene silencing and recapitulate large domains of PcG-dependent chromatin modifications, we investigated whether the resulting repressive chromatin would persist through cell divisions after release of the initiator. To determine if PcG protein targeting and histone modifications could be transmitted through mitotic cell divisions after reversal, we released TetR PcG protein fusion binding from TetO sites by adding Dox[25,26]. We treated the TetO-mESC lines with Dox for

6 days to allow approximately 10–12 cycles of replication and cell division based on the monitored growth rate (Supplementary Fig. 5a). This time interval would be sufficient to dilute any chromatin modifications to base line that are not maintained in the absence of the TetR fusion stimulus.

Release of TetR-Rybp by Dox resulted in rapid reactivation of GFP and BFP reporters as measured by flow cytometry (Fig. 2a). Dox had no effect on reporter gene expression in parental reporter cells, indicating a specific response to lack of vPRC1 (Supplementary Fig. 5b). ChIP analysis revealed that loss of silencing reflected displacement of vPRC1 and PRC2, and concomitant loss of their respective histone modifications, from 7xTetO as well as flanking regions (Fig. 2b). Hence, vPRC1-dependent chromatin modifications and gene silencing were not transmitted through genome replication and cell division in the absence of sequence-dependent TetR-Rybp recruitment.

In sharp contrast, Dox-dependent release of TetR-Cbx7 gave rise to a bimodal cell population: a small fraction of cells reactivated the reporter genes, yet the majority continued to silence GFP and BFP (Fig. 2c). This maintenance of reporter silencing was reproducible between independent clones and in a population of cells with different expression levels of TetR-Cbx7 (Supplementary Fig. 5c, d). In addition, sorting of GFP-positive and GFP-negative cells following Dox treatment demonstrated that the bimodal distribution reflects two separate populations (Supplementary Fig. 5e).

ChIP analysis after Dox treatment revealed efficient displacement of the TetR-Cbx7 from the 7xTetO site. However, TetR-Cbx7 was still enriched at flanking regions, co-localizing with Ring1B, Suz12 and repressive histone modifications (Fig. 2d). This redistribution of TetR-Cbx7-containing cPRC1 to regions flanking the 7xTetO site suggests a sequence-independent mechanism of propagation potentially linked to interaction with H3K27me3.

To determine if transmission of TetR-Cbx7-dependent repression through cell divisions was limited to the tandem reporter design or its genomic integration site, we generated three additional TetO-mESC lines by inserting a 7xTetO sequence with a single GFP reporter gene on chromosomes 1, 7, and 15; at loci devoid of transcriptional activity and PcG-dependent histone modifications in the parental cell line (Supplementary Tables 1 and 4). As in the original reporter line, expression of TetR-Cbx7 and -Rybp induced reporter gene silencing, yet maintenance of repression after Dox treatment was only observed in case of TetR-Cbx7 (Supplementary Fig. 6a–f).

One potential concern is that maintenance of PcG-dependent chromatin modifications and reporter gene silencing could arise from low affinity TetR binding to the DNA in the presence of Dox. To rule out a potential contribution of sequence-dependent initiation to the observed inheritance of repression, we sought to release Cbx7 tethering by induced genetic deletion of the TetR DNA binding domain. Genetic reversal was achieved by transducing TetO-mESCs with *TetR-Cbx7* or *TetR-Rybp* transgenes in which the sequence encoding the TetR DNA binding domain was fused to mCherry and flanked by loxP sites to enable tracking of Cre recombinase-mediated excision (Supplementary Fig. 7a). As expected, TetR-dependent recruitment of Cbx7 and Rybp resulted in reporter gene silencing (Fig. 2e, f). Following Cre recombinase transfection, mCherry-negative cells were isolated by FACS and precise TetR domain deletion was confirmed by Western blot (Supplementary Figs. 7b, c and 8). Importantly, after genetic deletion of the TetR DNA binding domain, flow cytometry confirmed selective maintenance of reporter gene silencing in cPRC1-mESCs but not vPRC1-mESCs (Fig. 2e, f and Supplementary Fig. 7d). Similar to Dox-dependent release, reversal of Cbx7 tethering by genetic TetR domain deletion

resulted in a bimodal population with the majority of cells maintaining reporter gene repression after 10–12 cell divisions (Supplementary Fig. 7e).

Hence, our direct comparison of reversible tethering of variant and cPRC1 complexes revealed striking differences in the heritable transmission of Polycomb-dependent repression. Unlike vPRC1, cPRC1 promotes sequence-independent maintenance of PcG-dependent gene repression.

**cPRC1 and PRC2 are required for heritable gene silencing**. To delineate the requirements for maintaining transcriptional gene silencing, we used CRISPR/Cas9 to generate loss-of-function (LOF) alleles in genes encoding different PRC1 and PRC2 components. We engineered cPRC1-TetO mESCs stably expressing Cas9 and validated sgRNAs targeting *Cbx7* (cPRC1), *Ring1B* (cPRC1, vPRC1), *Rybp* (vPRC1), or *Suz12* (PRC2), and confirmed protein loss by Western blot (Supplementary Figs. 9a, 10a and Supplementary Tables 2, 4). sgRNAs against an unrelated gene (*Slc6a6*) served as control. As expected, the initiation of GFP reporter silencing was lost in *Cbx7* LOF cells, consistent with loss of TetR-Cbx7 expression (Fig. 3a and Supplementary Fig. 9a, b). In contrast, reporter silencing initiation was unchanged in *Rybp* LOF and in control LOF cells. Upon Dox treatment, more than 60% of *Rybp* LOF cells remained GFP-negative (Fig. 3a and Supplementary Fig. 9b), similar to controls. These findings suggest that vPRC1 is not required to maintain the repression established by cPRC1. In the absence of Dox, *Ring1B* LOF had no impact on GFP silencing (Fig. 3a and Supplementary Fig. 9a, b). In these cells, Cbx7 and H2AK119ub1 levels were largely unaffected at the target locus, albeit slightly reduced (Supplementary Fig. 9c), suggesting initiation of Polycomb-dependent repression despite Ring1B loss. Indeed, Ring1A was upregulated and associated with TetR-Cbx7 and Mel18 (Supplementary Figs. 9d, e and 10b, c) suggesting at least partial functional compensation in the absence of Ring1B and consistent with previous reports[33]. In contrast, Dox treatment reactivated reporter gene expression in *Ring1B* LOF cells. Thus, maintenance of cPRC1-initiated repression relies on the functional integrity of endogenous cPRC1. Since Ring1B is responsible for much of the global H2AK119ub1 in mESCs[33] (Supplementary Fig. 9d), we speculate that cPRC1 function is nevertheless compromised in Ring1B LOF despite the upregulation of Ring1A.

Although TetR-Cbx7 was depleted at the 7xTetO site upon Dox treatment, it was still enriched at flanking regions, co-localizing with Suz12 and H3K27me3 (Fig. 2d). To assess the role of PRC2 and H3K27me3 in cPRC1 retention, we deleted PRC2 core subunit Suz12. CRISPR LOF mutation in *Suz12* revealed that functional PRC2 was required to maintain, but not initiate, repression (Fig. 3a and Supplementary Fig. 9a, b). As Suz12-deficient clones fail to assemble PRC2 and catalyze H3K27me3[34] (Supplementary Fig. 9c), these data suggest that PRC2 integrity and/or H3K27me3 are critical for sequence-independent cPRC1 targeting in mitotic cells.

**Histone modifications promote an epigenetic feedback loop**. To determine if heritable gene silencing requires H3K27me3, we utilized a selective inhibitor of the histone methyltransferase Ezh2, GSK126 (Ezh2i), which blocks catalytic activity dose-dependently without affecting the overall integrity of the PRC2 complex[35]. As expected, H3K27me3 was undetectable in Ezh2i-treated parental mESCs (Supplementary Fig. 11a). Importantly, inhibitor treatment did not disrupt initiation of GFP reporter silencing by TetR-Cbx7. In contrast, maintenance of reporter gene silencing was reduced by Ezh2i-treatment in a dose-dependent manner (Fig. 3b). Further, the Ezh2i-mediated

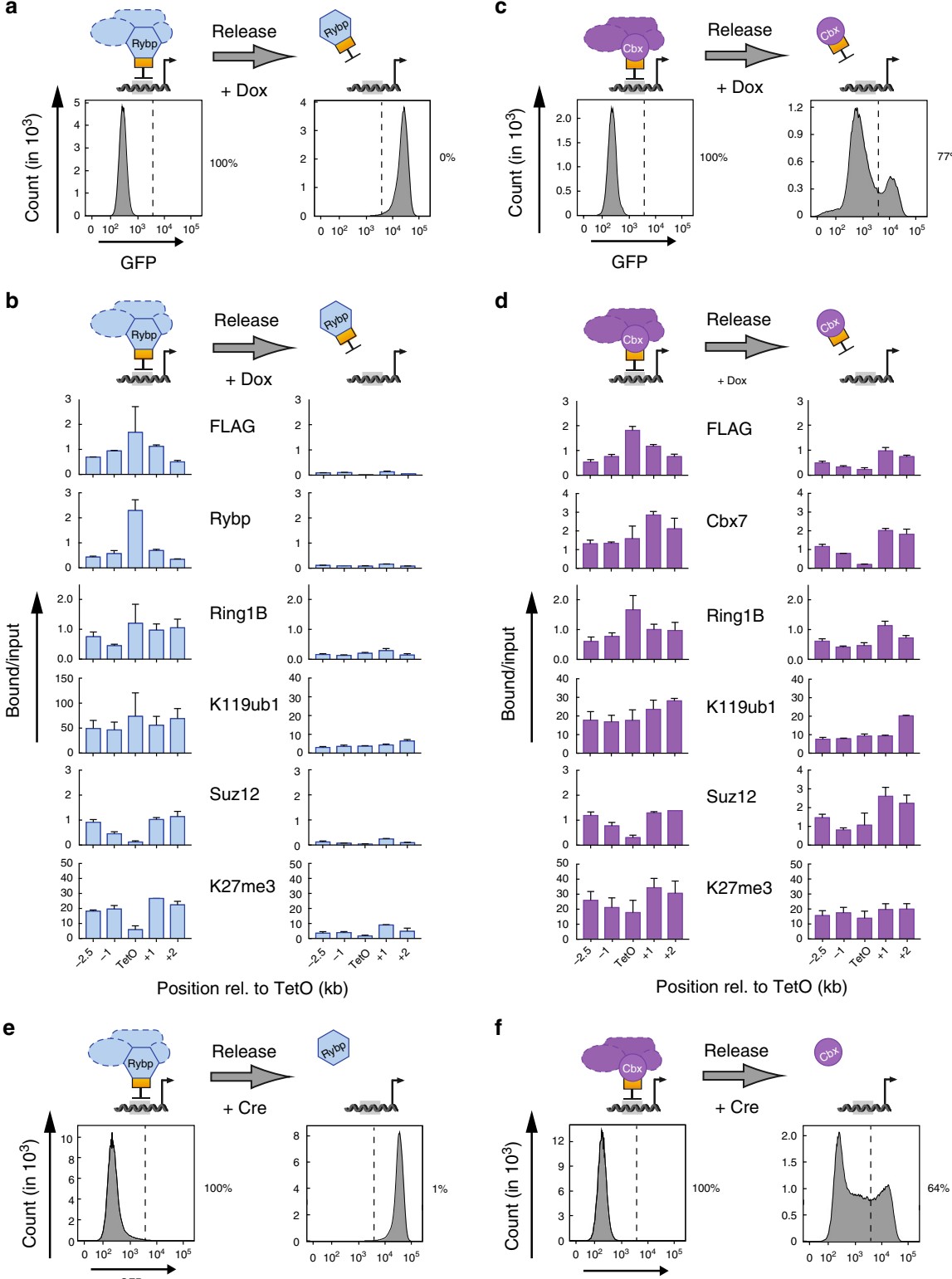

**Fig. 2** cPRC1 but not vPRC1 supports propagation of repressive chromatin. **a**, **c** Flow cytometry histograms relate GFP expression before and after release of TetR fusion protein recruitment in response to Dox treatment for 6 days. Percentages (%) indicate fraction of silenced cells. **b**, **d** ChIP qPCR analyses comparing relative enrichments of FLAG-TetR fusion proteins, PcG proteins and histone modifications before and after 6 days of Dox treatment. ChIP enrichments for H2AK119ub1 and H3K27me3 are normalized to negative control locus (IAP). Data are mean ± SD (error bars) of at least two independent experiments. Source data are provided as a Source Data file. **e**, **f** Flow cytometry histograms relate GFP expression before and 6 days after genetic release of Rybp and Cbx7 tethering by Cre-mediated deletion of the TetR DNA binding domain. Percentages (%) indicate fraction of silenced cells

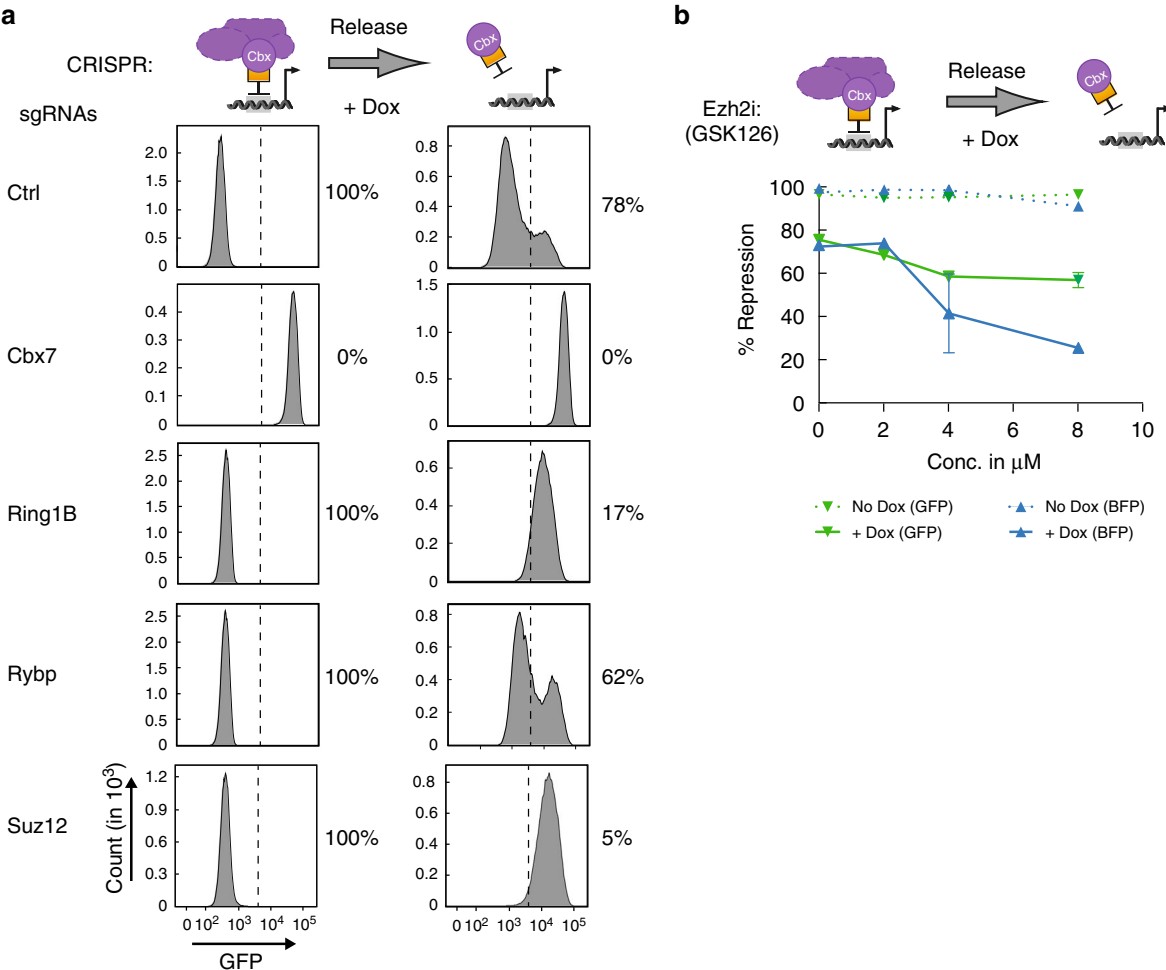

**Fig. 3** cPRC1 and H3K27me3 are required for maintenance of reporter gene silencing. **a** Flow cytometry histograms before and after 6 days of Dox treatment of CRISPR mutant clones isolated from sgRNA-treated cPRC1-mESCs. Percentage indicates fraction of GFP-negative reporter cells. **b** Percentage of GFP- and BFP-negative cells before and after 6 days of Dox treatment in response to increasing concentrations of Ezh2 inhibition by GSK126. Data are mean ± SD (error bars) of two independent experiments

reduction in silencing was more prominent at the distal BFP reporter than the proximal GFP promoter. Thus, H3K27me3 promotes heritable propagation of cPRC1 targeting in *cis* (Fig. 3b).

The chromodomain of Cbx7 has affinity for H3K9me3 and H3K27me3[9]. To determine if this interaction is involved in maintenance of reporter gene repression, we transduced TetO-mESCs with a TetR-Cbx7 mutant encoding a LOF amino acid substitution in the aromatic cage responsible for binding methylated histone (Cbx7[wt]/TetR-Cbx7[W35A])[9]. In addition, to discriminate the contribution of the endogenous wild-type allele, we expressed the TetR-Cbx7[W35A] mutant in TetO-mESCs with LOF mutation in the endogenous *Cbx7* gene (Cbx7[KO]/TetR-Cbx7[W35A], Supplementary Figs. 11b and 12b). TetR-Cbx7[W35A] was sufficient to nucleate functional cPRC1 and initiate repression of the dual reporter locus in wild-type and Cbx7[KO] TetO-mESCs (Fig. 4a, b and Supplementary Figs. 1b and 11c). However, unlike wild-type TetR-Cbx7, there was a small, yet significant fraction of reporter cells that escaped repression in response to recruitment of the Cbx7 cage mutant. This effect was more pronounced at the upstream BFP reporter, suggesting that the Cbx7 interaction with H3K27me3 contributes to the establishment of cPRC1-dependent gene silencing (Supplementary Fig. 11c). Indeed, ChIP analysis showed significantly less enrichment of Cbx7 and Ring1B at regions flanking the TetO site,

consistent with compromised cPRC1 binding to H3K27me3 in TetR-Cbx7[W35A] expressing cells (Fig. 4a).

Unexpectedly, both wild-type and Cbx7[KO] reporter cells expressing TetR-Cbx7[W35A] failed to maintain repression of GFP and BFP upon addition of Dox (Fig. 4b and Supplementary Fig. 11c). These data suggest that the contribution of endogenous Cbx7 to cPRC1-dependent reporter gene silencing in wild-type cells is limited. Indeed, we noticed that the expression of endogenous Cbx7 in cPRC1-mESCs was strongly downregulated, providing a potential explanation for the dominant impact of ectopic wild-type or mutant TetR-Cbx7 proteins in cPRC1 function (Supplementary Fig. 11b). To test if ectopic expression of wild-type Cbx7 would restore maintenance of reporter gene silencing, we introduced untagged Cbx7 in TetO-mESCs expressing the mutant TetR fusion (Cbx7[wt]/TetR-Cbx7[W35A]). However, similar to the reduction of endogenous protein, we did not observe additional expression of untagged Cbx7 in cPRC1-mESCs.

Given the reciprocal relationship between TetR fusion protein levels and endogenous Cbx7 expression, we sought to restore endogenous Cbx7 levels by eliminating the TetR-Cbx7 protein after initiation of reporter gene repression. Auxin-based degron systems enable inducible, proteasome-mediated destruction of target proteins[36]. TetR-Cbx7 fused to the Auxin-inducible degron (TetR-AID-Cbx7) and F-box protein TIR1 were transduced into

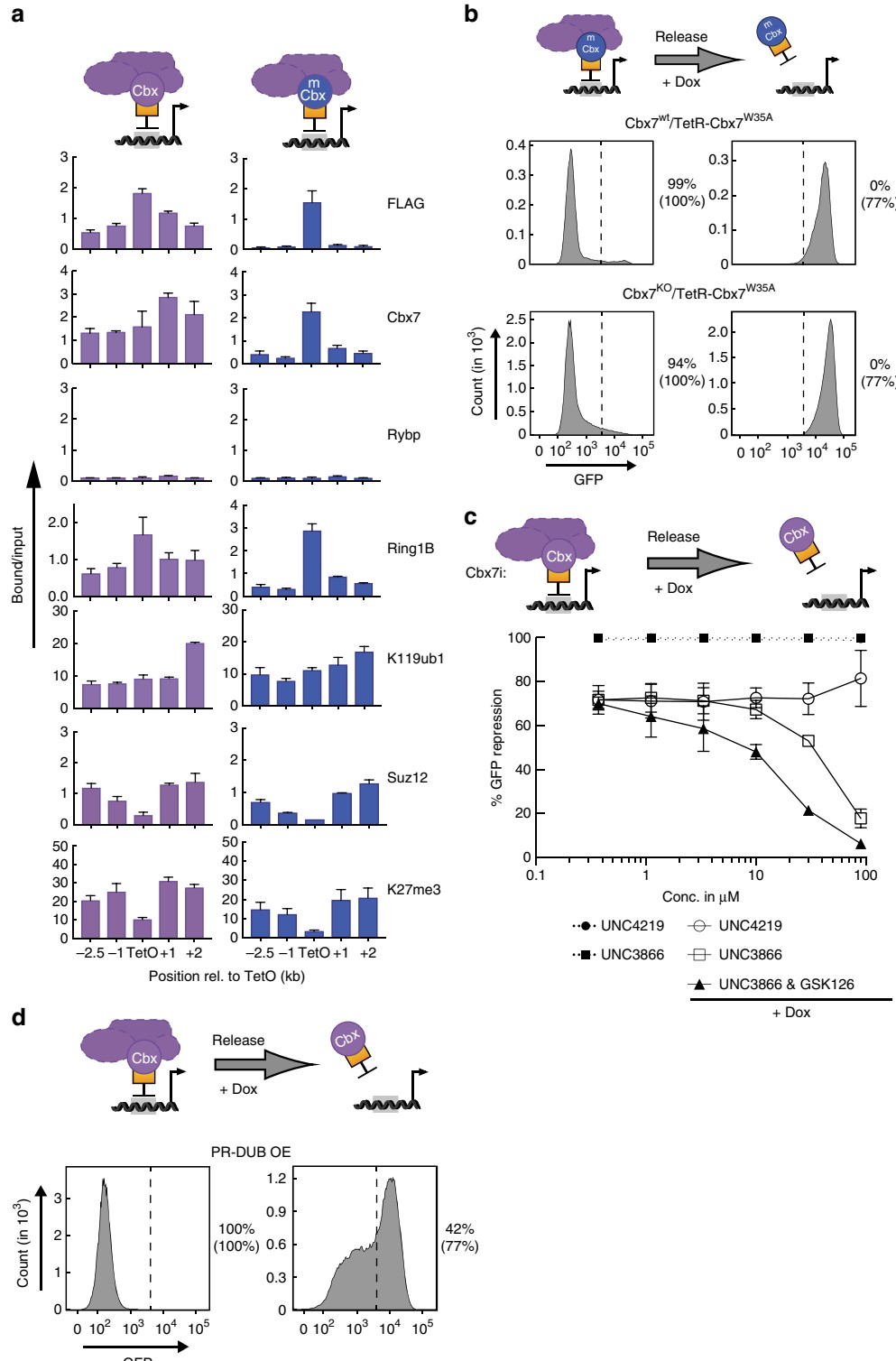

**Fig. 4** Interaction of Cbx7 with H3K27me3 is essential for cPRC1-dependent inheritance. **a** ChIP qPCR analyses compares relative enrichments of PcG proteins and histone modifications in reporter cells expressing wild-type and mutant FLAG-TetR-Cbx7. ChIP enrichments for H2AK119ub1 and H3K27me3 are normalized to negative control locus (IAP). Data are mean ± SD (error bars) of at least two independent experiments. Source data are provided as a Source Data file. **b** Flow cytometry histograms compare GFP expression before and after three days of Dox treatment of wild-type (gray—upper panels) and Cbx7$^{KO}$ dual reporter cells expressing TetR-Cbx7$^{W35A}$ (gray—lower panels). Percentages indicate fraction of silenced reporter cells expressing either mutant or wild-type TetR fusion (in brackets). **c** Percentage of GFP-negative cells before and after 6 days of Dox treatment in response to increasing concentrations of Cbx7 inhibitor (UNC3866) alone, in combination with 4 μM GSK126 or control compound (UNC4219). Data are mean ± SD (error bars) of two independent experiments. **d** GFP histograms before and after 6 days of Dox addition to TetR-Cbx7 reporter cells with overexpression of Bap1 and Asxl1 (PR-DUB OE), components of the human PR-DUB complex specific for H2AK119ub1. Percentages indicate fraction of silenced cells in PR-DUB OE and wild-type reporter cell lines (in brackets)

TetO-mESCs as well as Cbx7[KO] TetO-mESCs, and clones with high TIR1 expression were isolated (Supplementary Figs. 11d and 12b). As expected, expression of TetR-AID-Cbx7 led to silencing of GFP and BFP reporters in both cell lines. Silencing was largely maintained in the presence of Dox treatment for three days (Supplementary Fig. 11e). To discern the contribution of ectopic versus endogenous Cbx7, we treated with Auxin to degrade TetR-AID-Cbx7. Western blot analysis revealed Auxin-dependent degradation of the TetR fusion protein in both reporter cell lines (Supplementary Fig. 11d). Importantly, endogenous Cbx7 levels in wild-type TetO-mESCs were restored to 40% of the Cbx7 levels observed in parental TetO-mESCs levels (Supplementary Fig. 11d). To ensure complete displacement and TetR fusion protein degradation, we subjected reporter cells to combined Auxin and Dox treatment, which was well-tolerated. After three days of Auxin/Dox treatment, repression of both reporter genes was lost in more than 90% of Cbx7[KO] TetO-mESCs (Supplementary Fig. 11e). In contrast, nearly 40% of reporter cells remained GFP-negative in wild-type TetO-mESCs, suggesting that low levels of endogenous Cbx7 can contribute to the propagation of cPRC1-dependent repression. Importantly, Cbx7 levels are critical as ectopic expression further enhanced faithful propagation of reporter gene repression. Together, these results suggest that Cbx7 binding to methylated histones via its chromodomain is essential for propagation of Polycomb domains and gene silencing.

Our results above revealed that ectopic expression of TetR-Cbx7 influences the activity and levels of endogenous Cbx7. In contrast, ectopic expression of TetR-AID-Rybp did not affect the levels of endogenous Cbx7 or Rybp (Supplementary Figs. 11f and 12c). TetR-AID-Rybp led to reporter gene silencing in TetO-mESCs, as expected (Supplementary Fig. 11g). Further, Auxin treatment reduced the levels of TetR-AID-Rybp and reactivated GFP and BFP within 3 days (Supplementary Fig. 11g), consistent with our results above (Fig. 2). These data confirm that cPRC1-initiated but not vPRC1-initiated silencing is maintained upon elimination of the initial stimulus, and suggest that H3K27me3 enrichment is necessary but not sufficient to maintain silencing by endogenous Cbx7.

To validate the requirement for the interaction between Cbx7 and H3K27me3, we utilized the PRC1 inhibitor UNC3866 (Cbx7i), which selectively binds to Cbx4/7 and disrupts the interaction with methylated histones[37]. cPRC1-TetO mESCs were treated with either the Cbx7 antagonist UNC3866 or the negative control compound UNC4219, both in absence and presence of Dox. Neither compound had a significant effect on the initiation of reporter gene silencing (Fig. 4c and Supplementary Fig. 13a, b). In the presence of Dox, maintenance of repression was disrupted only by Cbx7i treatment in a dose-dependent manner. In contrast, >70% of reporter cells treated with the control compound maintained GFP- and BFP-repression. In line with Cbx7 interacting specifically with H3K27me3, combined treatment with Cbx7i and Ezh2i exacerbated the failure to maintain gene repression (Fig. 4c and Supplementary Fig. 13a, b), in agreement with the conclusion that persistent repression relies on H3K27me3 binding. Highlighting the reversible nature of heritable cPRC1-dependent repression, these results support the conclusion that cPRC1 can promote sequence-independent maintenance of PcG-dependent gene silencing. Moreover, disruption of the "reader function" of Cbx7, either via mutation or pharmacologic antagonism, abrogates epigenetic targeting of cPRC1 after genome replication in mitotic cells.

Our results suggest that PRC1 can promote PRC2 targeting (Fig. 1c), consistent with recent reports[30,38]. Indeed, Jarid2, an auxiliary component of PRC2, was recently shown to bind H2AK119ub1, the modification generated by PRC1[32].

H2AK119ub1 might also contribute to silencing of gene expression[6,7]. To investigate the role of H2AK119ub1 in initiation and propagation of cPRC1-mediated gene repression, we ectopically expressed Bap1 together with a truncated version of Asxl1 (1–479 aa) to generate a hyperactive Polycomb Repressive-Deubiquitinase complex (PR-DUB) in cPRC1-TetO mESCs[39] (Fig. 4d). Western blot analysis confirmed that PR-DUB overexpression was sufficient to reduce total levels of H2AK119ub1, similarly to a recent report[39] (Supplementary Figs. 13c and 14). While bulk H2AK119ub1 reduction had a negligible effect on the initiation of gene silencing it significantly reduced maintenance of gene silencing (Fig. 4d and Supplementary Fig. 13d), suggesting that H2AK119ub1 is important to recruit PRC2 and/or cPRC1 for epigenetic maintenance of cPRC1-initiated gene silencing.

## Discussion

We have uncovered previously unappreciated, non-redundant functions of the major vertebrate PRC1 complexes in the propagation of gene silencing. We demonstrate that Polycomb-dependent heritable gene repression is restricted to regulation by cPRC1. In *Drosophila*, it was shown that reversible targeting of Polycomb complexes, including the Cbx homolog Pc, was sufficient to establish repressive chromatin modifications and initiate gene silencing[20,21,40]. However, whether propagation of repression in the fruit fly requires continuous sequence-dependent recruitment of PcG proteins to Polycomb response elements (PREs) within DNA is controversial. Similarly, the emerging contribution of unmethylated CpG islands in directing PRC1 and PRC2 to target genes[17,41,42] challenges the model of sequence-independent targeting mechanisms involved in inheritance of Polycomb-dependent gene repression in vertebrates. To address these challenging questions in basic epigenetics research, we have taken a reductionist approach enabling reversible, sequence-specific initiation of cPRC1- and vPRC1-dependent chromatin modifications at a synthetic reporter locus in mouse ES cells and evaluated their capacity to self-propagate through cycles of DNA replication and cell divisions in the absence of the initial stimulus. Our results suggest that in vertebrates Polycomb-repressive domains can be propagated by a selective feedback mechanism between cPRC1 and PRC2 which is sustained by *cis*-acting histone modifications (Fig. 5a, b).

Using the TetOFF system, we show that cPRC1 is sufficient to initiate a repressive Polycomb chromatin domain which can promote sustained gene silencing through DNA replication and cell divisions even after release of the initial stimulus. cPRC1-mediated gene silencing persisted not only after Dox-dependent release but also after genetic TetR reversal arguing for a sequence-independent mechanism of inheritance. While the duration of inheritance varied between chemical and genetic release of tethering, either case resulted in a bimodal population with the majority of cells maintaining the OFF state for more than 6 days (10–12 cell divisions). We find that propagation of Cbx7-initiated repression involves H3K27me3 binding which can be abrogated by genetic or chemical PRC2 inactivation or by blocking of CBX methyl-lysine recognition domain (Fig. 5a). Together, sequence-independence and the reversible nature of cPRC1-dependent propagation of gene silencing strongly argue for an epigenetic mechanism.

Interestingly, our reductionist system reveals that propagation of cPRC1-dependent repression is sensitive to cellular Cbx7 abundance. In mESCs, endogenous levels of Cbx7 promoted only limited maintenance of gene silencing, yet this was significantly enhanced by additional ectopic expression. Elevated Cbx7 levels may favor cPRC1 assembly promoting maintenance of

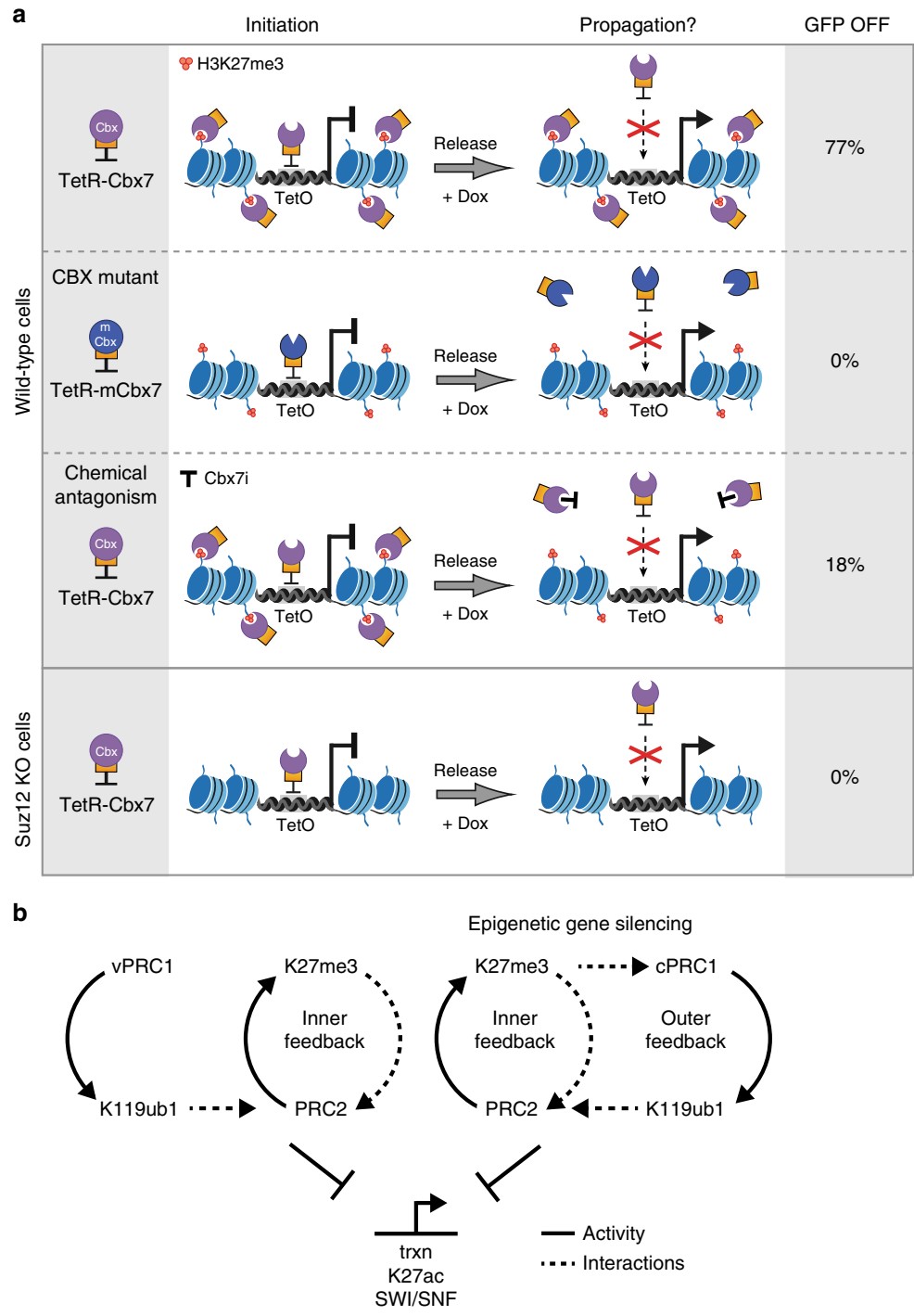

**Fig. 5** Summary of key experiments and model of differential epigenetic regulation by cPRC1 and vPRC1. **a** Schematic representation summarizes the key experiments in support of an interaction between the CBX methyl-lysine recognition domain and H3K27me3 promoting sequence-independent, heritable propagation of cPRC1 recruitment and gene silencing. Results of tethering and release of TetR-Cbx7 or TetR-Cbx7[W35A] (TetR-mCbx7) in wild-type and Suz12 LOF TetO-mESCs. Chemical antagonism indicates treatment of TetR-Cbx7 reporter cells with Cbx7i (UNC3866). GFP OFF displays fraction of transcriptionally silent reporter cells. **b** Model of the regulation of canonical and variant PRC1 complexes in relation to PRC2. Only cPRC1 and PRC2 engage in a reciprocal feedback mechanism to promote sequence-independent epigenetic gene silencing. Arrows indicate catalytic activity of different PcG complexes. Dashed arrows indicate interactions with histone modifications. Stop bars highlight antagonistic effects

target-gene repression through enhanced H3K27me3 interaction. To this extent, it remains to be shown if propagation of Polycomb-dependent silencing is also enhanced in cell-types with high endogenous expression of Cbx7 or its Polycomb paralogs. Nevertheless, we find that silencing was sustained by cPRC1 and

PRC2 interactions through *cis*-acting histone modifications arguing that ectopic, reversible Cbx7 tethering triggers a physiological sequence-independent feedback mechanism for PcG protein recruitment in mESCs. Hence, despite the synthetic nature of establishment, our findings provide proof-of-principle

and reveal a potential mechanism of heritable Polycomb-dependent repression in vertebrates.

While a positive feedback cycle between PRC2 and its catalytic output H3K27me3 has been proposed previously[13,14], our results suggest that in the context of transcriptional antagonism, self-reinforcement of H3K27me3 requires additional feedback by CBX methyl-lysine recognition dependent cPRC1 activity, ensuring robust repression and installation of H2AK119ub1 to promote further PRC2 recruitment. Similar to its canonical counterpart, repressive chromatin-modifying activities of vPRC1 can induce gene silencing. However, without the ability to "read" H3K27me3, vPRC1 may not contribute to epigenetic propagation of repression (Fig. 5b). The substantial overlap in genomic binding[24] between the different PRC1 complexes suggests that they might act sequentially at target loci whereby transient repression initiated by vPRC1 is converted into long-term heritable gene silencing by concerted action of PRC2 and cPRC1. Alternatively, vPRC1 may enforce a more dynamic mode of gene repression at its target genes. Since despite high levels of H3K27me3, we observe very little Cbx7 and Cbx2 binding in TetR-Rybp expressing reporter cells (Fig. 1c and Supplementary Fig. 13e), vPRC1 may establish repressive chromatin modifications that potentially evade cPRC1 targeting and long-term propagation. We propose that the diversification of PRC1 complexes has allowed vertebrates to evolve a large repertoire of chromatin regulatory mechanisms for fine-tuning gene repression in response to the increased complexity of intrinsic and extrinsic stimuli.

## Methods

**Construct design and delivery**. All constructs were created as lentiviral plasmids expressing the gene of interest linked to mCherry CDS under the control of an EF1a- or an UCOE-SFFV promoter. Lentivirus was produced by polyethylenimine (PEI) co-transfection of the desired construct and two packaging vectors VSV-G (Addgene #8454) and psPAX2 (Addgene #12260) in Lenti X 293T cells (Takara #632180). After 48–72 h, the virus was collected. mESCs were then transduced with the virus for 48 h in the presence of 8 μg/ml polybrene (Santa Cruz Biotechnology, SACSC-134220).

**Generation and culture conditions of mESCs**. All mESCs used in this study were derived from haploid mESCs available at Haplobank repository[43]. TetO mESCs were cultivated without feeders in high-glucose-DMEM supplemented with 13.5% fetal bovine serum (Sigma), 10 mM HEPES pH 7.4, 2 mM GlutaMAX (Gibco), 1 mM sodium pyruvate (Sigma), 100 U penicillin/ml (Sigma), 0.1 mg streptomycin/ml (Sigma), 1× non-essential amino acids (Sigma), 50 μM beta-mercaptoethanol (Gibco) and recombinant LIF (37 °C, 5% CO$_2$). TetO-mESCs with BFP and GFP reporters were generated by recombinase-mediated cassette exchange introducing the reporter DNA YR06 (Fig. 1b and Supplementary Table 1) into a gene-trap located on chromosome 15[44]. TetO-mESCs with single GFP reporter were generated by introducing the reporter DNA HFM93 on chromosome 1, the reporter DNA HFM91 on chromosome 7 and the reporter DNA HFM92 on chromosome 15 via CRISPR/Cas9 assisted homologous recombination. Reversal of TetR fusion protein binding was achieved by addition of 1 μg/ml Doxycycline (final - Sigma, D9891) to mES cell culture medium.

**Flow cytometry analysis and sorting**. All flow cytometry analyses were conducted on a LSR Fortessa (BD Biosciences) using BD FACS Diva or FlowJo software. For fluorescent cell sorting a FACS ARIA III (BD Biosciences) was used. Isolation of haploid mESCs entailed incubation with 20 μg/ml Hoechst 33342 (Thermo Scientific Fisher) for 30 min at 37 °C and 5 % CO$_2$ prior to FACS. Selection of transgene expression by Thy 1.1 required surface staining with a Thy1.1 specific antibody. After incubation in PBS containing 1 % FBS with Fc-blocking antibody at 1:500 (Affymetrix eBioscience Anti-Mouse CD16/CD32 Purified) for 5 min at RT, mESCs were treated with Thy 1.1 antibody (Affymetrix eBioscience Anti-Mouse/Rat CD90.1 (thy-1.1) APC-eFluor 780) at 1:750 for 30 min. For the Flow cytometry-based strategy for the isolation and of wild-type and CRISPR mutant cPRC1- and vPRC1-TetO-mESCs, please see Supplementary Fig. 3.

**Chromatin immunoprecipitation (ChIP-qPCR)**. For Chromatin Immunoprecipitation, 30–50 × 10$^6$ mESCs were trypsinized for 6–8 min prior to quenching with FBS containing ES cell medium. 25 × 10$^6$ mES cells were collected, washed in once

in 1× PBS and crosslinked with formaldehyde at a final concentration of 1% for 7 min. The crosslinking was stopped on ice and with glycine at final 0.125 M concentration. The crosslinked cells were pelleted by centrifugation for 5 min at 1200 × g at 4 °C. Nuclei were prepared by washes with NP-Rinse buffer 1 (final: 10 mM Tris pH 8.0, 10 mM EDTA pH 8.0, 0.5 mM EGTA, 0.25% Triton X-100) followed by NP-Rinse buffer 2 (final: 10 mM Tris pH 8.0, 1 mM EDTA, 0.5 mM EGTA, 200 mM NaCl). Afterwards the cells were prepared for shearing by sonication by two washes with Covaris shearing buffer (final: 1 mM EDTA pH 8.0, 10 mM Tris-HCl pH 8.0, 0.1% SDS) and resuspension of the nuclei in 0.9 mL Covaris shearing buffer (with 1 × protease inhibitors complete mini (Roche)). The nuclei were sonicated for 15 min (Duty factor 5.0; PIP 140.0; Cycles per Burst 200; at 4 °C) in 1 ml Covaris glass cap tubes using a Covaris E220 High Performance Focused Ultrasonicator. Input samples were prepared from 25 μL sonicated lysate. Therefore, chromatin was RNase A and Proteinase K digested and crosslink reversed overnight at 65 °C. DNA was then precipitated and shearing of DNA was confirmed to be between 500 and 1000 bp by agarose gel electrophoresis. Crude chromatin lysate was further processed by spinning at 20,000 × g at 4 °C for 15 min and supernatant used for ChIP. An equivalent of 50 μg DNA was incubated overnight in 1× IP buffer (final: 50 mM HEPES/KOH pH 7.5, 300 mM NaCl, 1 mM EDTA, 1% Triton X-100, 0.1% DOC, 0.1% SDS) with following antibodies at 4 °C on a rotating wheel: 0.5 μl H3K27me3 (Diagenode, C15410195), 3 μl Ring1B (Cell Signaling, D22F2), 1.5 μl Suz12 (Cell Signaling, D39F6), 1.5 μl H3K4me3 (Millipore, 05–745R), 7.5 μl Mel18 (Santa Cruz, sc-10744), 2 μl Cbx7 (Abcam, ab21873), 1.5 μl RYBP (Sigma Aldrich, PRS2227), 1.5 μl FLAG (Sigma Aldrich, F1804), 1.5 μl H3K27ac (Abcam, ab4729), 1.5 μl H2AK119ub (Cell Signaling, D27C4), 7.5 μl Gal4 (Santa Cruz, sc-510). The overnight IPs were incubated with BSA-preblocked Protein G coupled Dynabeads (Thermo Fisher Scientific) for more than 6 h at 4 °C on a rotating wheel. IPs were subsequently washed 5× with 1× IP buffer (final: 50 mM HEPES/KOH pH 7.5, 300 mM NaCl, I mM EDTA, 1% Triton-X100, 0.1% DOC, 0.1% SDS), 3× with DOC buffer (10 mM Tris pH 8, 0.25 mM LiCl, 1 mM EDTA, 0.5% NP40, 0.5% DOC) and 1× with TE (+50 mM NaCl). The DNA was then eluted 2× with 150 μL Elution buffer (final: 1% SDS, 0.1 M NaHCO$_3$) for 20 min each at 65 °C. The eluate was treated with RNase A and Proteinase K and crosslink reversed overnight at 65 °C. The IP DNA was PCIA extracted and precipitated and quantified using qPCR on a CFX Connect Real-Time PCR Detection System (Biorad). qPCR primers are listed in Supplementary Table 3.

**Western blot**. Nuclear extract from 10 × 10$^6$ mESCs was obtained by lysis in Buffer A (final: 25 mM Hepes pH 7.6, 5 mM MgCl$_2$, 25 mM KCl, 0.05 mM EDTA, 10% Glycerol, 1 mM DTT, 1 mM PMSF, 1× Complete Mini protease inhibitor) followed by collection in RIPA buffer (final: 150 mM NaCl, 1% triton, 0.5% sodium deoxy-cholate, 0.1% SDS, 50 mM Tris pH 8.0). Nuclear extracts were homogenized by sonication in a Diagenode Bioruptor and concentration was determined by Bradford assay (Biorad). 20 μg/lane total protein was run on Novex Life Technology NuPAGE 4–12% Bis-Tris gels in Invitrogen NuPAGE MES SDS Running Buffer and transferred on a Merck Chemicals Immobilon-P Membrane (PVDF 45 μm). The membrane was blocked (5% non-fat dry milk in 1× PBS, 0.1% Tween 20) and incubated in 5% non-fat dry milk in 1× PBS and 0.1% Tween 20 with the primary antibodies as listed in Supplementary Table 2. Finally, the membrane was incubated with corresponding secondary HRP coupled antibodies (5% non-fat dry milk in 1× PBS, 0.1% Tween 20), developed using Clarity Western ECL Substrate (Biorad) and imaged by a ChemiDoc XRS+ Imaging system (Biorad). For uncropped Western blots with molecular weight markers, please see Supplementary Figs. 2, 8, 10, 12 and 14.

**Co-immunoprecipitation**. Whole cell protein extract from 45 × 10$^6$ mESCs was obtained by lysis in 500 μl Buffer B (final: 20 mM Tris-HCl pH 7.5, 150 mM NaCl, 2 mM MgCl$_2$, 10% Glycerol, 1 mM DTT, 1 mM PMSF, 0.2% NP-40, 1× Complete Mini protease inhibitor). Lysate was homogenized by sonication in a Diagenode Bioruptor followed by rotation for 3 h at 4 °C. After 30 min centrifugation at 4 °C protein concentration of the lysate was determined by Bradford assay (Biorad). In parallel, 30 μl Protein G coupled Dynabeads (Thermo Fisher Scientific) were prepared for each IP reaction as follows: 3x wash in Buffer B, incubation with 1.5 μg FLAG antibody (Sigma Aldrich, F1804) for 3 h at 4 °C, 1× wash in Buffer B and finally resuspension in 30 μl of Buffer B. For each IP, 30 μl of pre-bound Dynabeads were incubated with 3 mg protein extract in a final volume of 500 μl overnight at 4 °C. Finally, beads were washed four times with Buffer B and proteins were eluted at 95 °C in SDS sample buffer and analyzed by Western blots.

**CRISPR/Cas9 editing in cPRC1-TetO mESCs**. TetO-mESCs with LOF mutation in the endogenous *Cbx7* gene and LOF mutations of endogenous PcG genes in cPRC1-TetO mESCs were obtained by CRISPR/Cas9 technology. CRISPR guide RNAs were designed using the online tool of the Zhang lab (http://crispr.mit.edu, Zhang, MIT 2015) and cloned in modified lentiviral CRISPR/Cas9 expression vectors expressing the gRNAs (Supplementary Table 4) with improved tracer driven by a U6 promoter and a wild-type hSPCas9 with either a Thy1.1 marker or a blasticidin selection marker separated by a P2A driven by an EFS promoter (generous gift from J. Zuber). Parental TetO-mESCs were co-transfected with CRISPR/Cas9 expression vectors and a 200 bp double-stranded DNA

oligonucleotide with homology arms flanking a substitution of GCT for TGG using the Amaxa nucleofection protocol for mESCs (Lonza). After 24–36 h the cells were sorted positive for Thy1.1. Cbx7$^{KO}$ TetO-mESC clones were identified by Western blot analysis. CRISPR/Cas9 expression vectors to disrupt endogenous PcG genes was delivered into cPRC1-TetO mESCs via lentiviral infection followed by blasticidin selection.

**Chemical Inhibition of Ezh2 and/or Cbx4/7 in cPRC1-TetO mESCs.** $4 \times 10^3$ cPRC1-TetO mESCs were treated for three days on 96 well plates in both absence and presence of 1 µg/ml doxycycline (Sigma, D9891) with following chemical inhibitors: Ezh2 inhibitor GSK126 (Axora, BV-2282), increasing concentrations of negative control compound UNC4219, Cbx4/7 antagonist UNC3866 alone or in combination with 4 µM GSK126[37]. Dilutions of UNC3866 and UNC4219 were prepared in DMSO.

**Generation of mESCs growth curves.** $1 \times 10^5$ respective mESCs were plated in the beginning. After 24, 48, and 72 h mESCs were collected and stained with trypan blue for counting (Countess$^{TM}$, Invitrogen AMQAX1000). Cell counts were performed in duplicates.

**Conditional depletion of TetR-AID-Cbx7 and TetR-AID-Rybp.** Parental TetO-mESCs and Cbx7$^{KO}$ TetO-mESCs were transduced with DB52 and DB53 and clones with high TIR1 and TetR-AID-Cbx7 expression were isolated. Parental TetO-mESCs were transduced with DB53 and DB64 and clones with high TIR1 and TetR-AID-Rybp expression were isolated. All cells were treated for 72 h in the presence or absence of Doxycycline (1 µg/ml final concentration) alone or in combination with Indole-3-acetic acid sodium salt (Auxin) (Sigma, I5148–500 µM final concentration).

**Genetic deletions of TetR DNA binding domains.** TetO-mESCs were transduced with YR111 and YR112 and mCherry-positive clones were isolated. For genetic reversal of TetR fusion protein binding, reporter cell clones expressing conditional TetR fusions were transduced with Cre recombinase using the mouse ES Cell Nucleofector Kit (Lonza) and Thy1.1-positive cells were sorted after 24–36 h. Flow cytometry analysis of mCherry and GFP expression was carried out after 96 h. Both nuclear protein extracts and genomic DNA were collected reporter cells prior (mCherry-positive) and after (mCherry-negative) transfection with Cre recombinase.

**Reporting summary.** Further information on experimental design is available in the Nature Research Reporting Summary linked to this article.

## Data availability

All relevant data supporting the key findings of this study are available within the article and its Supplementary Information files or from the corresponding author upon reasonable request. Source data for graphs in Figs. 1c, 2b, d and 4a and Supplementary Figs. 1c and 9c are provided as a Supplementary Source Data file. A reporting summary for this Article is available as a Supplementary Information file.

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

## Acknowledgements

We thank the IMBA/IMP BioOptics facility and Molecular Biology Service, J. Jude, M. Muhar, and M. Aichinger for reagents and technical support, D. Schübeler, A. Stark, J. Brennecke, and M. Leeb for comments on the manuscript. We apologize to colleagues whose work could not be cited due to space limitations. This work was supported by the Austrian Academy of Sciences, the New Frontiers Group of the Austrian Academy of Sciences (NFG-05), and from the Human Frontiers Science Programme Career Development Award (CDA00036/2014-C). R.Y. has received funding from an EMBO Long-Term Postdoctoral Fellowship (EMBO ALTF 256–2015). J.A.M. was supported by Boehringer Ingelheim Fellowship. S.V.F., L.I.J., and J.I.S. have received funding from the National Institute of General Medical Sciences, U.S. National Institutes of Health (NIH, grant R01GM100919) and the University Cancer Research Fund, University of North Carolina at Chapel Hill. U.E. is supported by the City of Vienna (MA 7), the Federal Ministry of Science, Research and Economy and the Austrian Academy of Sciences.

## Author contributions

H.F.M. and O.B. initiated and designed the study. H.F.M., D.B., C.P., L.M., and R.Y. performed the experiments. R.Y. generated the dual reporter TetO mESC line; R.Y. and K.S. made the initial observation of Cbx7-dependent maintenance of reporter silencing. H.F.M. generated the single reporter TetO mESCs. K.B., J.A.M., and J.W. supported experiments and data analysis. J.I.S., L.I.J., and S.V.F. provided chemical probes and experimental advice. U.E. generated the parental mESCs and provided technical advice. O.B. supervised all aspects of the project; the manuscript was prepared by H.F.M., D.B. and O.B. All authors discussed results and commented on the manuscript.

## Additional information

**Competing interests:** The authors declare no competing interests.

