## [Peer Review File · Nature Communications]

REVIEWERS' COMMENTS:

Reviewer #1 (Remarks to the Author):

The latest revised version of the manuscript by Oliver Bell and colleagues titled "Canonical PRC1 controls sequence-independent propagation of Polycomb-dependent gene silencing" presents some new experiments in the context of the authors' system for reversibly tethering Polycomb proteins to a defined locus in mouse embryonic stem cells. Upon Polycomb protein tethering, the authors observe gene repression whose persistence after release of the tethered protein differs according to whether a member of canonical or variant PRC1 is used. The authors conclude from their results that canonical PRC1 establishes silencing that can be inherited in a sequence-independent fashion, whereas variant PRC1 does not.

A close examination of the previous version of the manuscript raised fundamental questions regarding the soundness of the conclusions due to major weaknesses in the experimental design. The authors describe efforts to address these issues in a Rebuttal.

First, it appeared that the tethered canonical PRC1 protein FLAG-TetR-Cbx7 remained bound to the TetO sites even after Dox-mediated release. The authors contend that the FLAG ChIP signal (now in Fig. 2d) indeed represents binding of the FLAG-TetR-Cbx7 protein to the TetO region and particularly to sites flanking the actual TetO sites, but that this binding is sequence-independent. In support of this interpretation, they note that the W35A chromodomain mutant FLAG-TetR-mCbx7 does not bind to these sites even prior to Dox addition. The problem with this explanation is not that it is not plausible, but rather that the alternative scenario of residual initiation is not ruled out experimentally. The release of the tethered protein by Cre-mediated excision of the TetR domain is potentially much more convincing, but in this case (comparing Fig. 2 panels c and f, and looking at Rebuttal Fig. 4d as well) it seems from the shape of the histogram that the repression decays much more rapidly, thereby undercutting the authors' message and casting doubt on all of the Dox-based experiments presented in the manuscript.

Second, the authors acknowledged that there was "only limited contribution of endogenous wild-type Cbx7" to the maintenance of gene repression after release of tethered Cbx7, implying that this maintenance depends critically on the FLAG-TetR-Cbx7 protein, proposed to act independently of its DNA-binding domain at this stage. The authors provide new evidence in support of this idea in Fig. 2f and Rebuttal Figs. 4 and 5, as a stabilized version of the FLAG-Cbx7 protein after TetR domain excision mediates longer persistence of GFP silencing. The authors' conclusion that endogenous Cbx7 is not sufficient to mediate propagation of gene silencing in this system is correct, but it significantly undermines the impact of the study.

The purpose of artificially perturbing biological systems is to learn about the mechanisms that govern the natural function of those systems. Here, the authors have described mechanisms that apply only to the artificial function of their engineered cell lines. Their findings help to explain how gene repression established via forced recruitment of overexpressed Cbx7 is propagated in a sequence-independent manner by the same overexpressed Cbx7, although as noted above any reasonable conclusions should be limited to the experiments employing release through TetR excision. Even allowing this much, however, it is not clear that the findings provide any insight that is relevant for understanding how inheritance of Polycomb-dependent silencing works endogenously. One might be tempted to say that the authors have uncovered an important mechanistic difference in the respective functions of canonical PRC1 component Cbx7 and variant PRC1 component Rybp, but then again one might also be tempted to attribute that difference more

trivially to the vastly different levels of expression of the TetR-FLAG fusion proteins as compared to their endogenous counterparts (Rebuttal Fig. 5a).

In short, because the Dox-mediated release assay that is at the heart of the study provides results that are demonstrably different from a more rigorous excision-based release method, and because even the conclusions that are well supported do not advance the field's understanding of Polycomb-mediated inheritance of gene repression, we cannot support publication of the manuscript.

Reviewer #2

(no comments for the authors)

Detailed point-by-point response:

Response to Reviewer #1

We thank this Reviewer for her/his feedback. She/he raised valuable concerns relating to the ectopic expression of TetR fusions and the displacement from the TetO binding site upon Dox treatment. We have addressed these concerns with textual edits in the revised version of the manuscript discussing the potential caveats of our synthetic approach.

Reviewer #1: "...the alternative scenario of residual initiation is not ruled out experimentally. The release of the tethered protein by Cre-mediated excision of the TetR domain is potentially much more convincing, but in this case (comparing Fig. 2 panels c and f, and looking at Rebuttal Fig. 4d as well) it seems from the shape of the histogram that the repression decays much more rapidly, thereby undercutting the authors' message and casting doubt on all of the Dox-based experiments presented in the manuscript. ..."

Response: We appreciate the reviewer's concern and have included a brief paragraph in the revised version of our manuscript discussing differences in maintenance of silencing in response to Dox-dependent and genetic reversal of tethering. Importantly, despite variable duration, either chemical or genetic release resulted in a bimodal population of which more than 60% maintained the OFF state for at least 10-12 cell divisions. In contrast, we never observed maintenance of Rybp-initiated repression.

Reviewer #1: "...The authors' conclusion that endogenous Cbx7 is not sufficient to mediate propagation of gene silencing in this system is correct, but it significantly undermines the impact of the study. ..."

Response: We agree with this reviewer that our experiments using induced degradation of FLAG-TetR-Cbx7-AID in wild-type mESCs revealed a significant but limited contribution of endogenous Cbx7 to maintenance of TetR-Cbx7-initiated silencing. However, despite the artificial stimulus, we show that silencing is sustained by *cis*-acting histone modifications, PRC2-mediated H3K27me3 and cPRC1-mediated H2AK119ub1, arguing ectopic, reversible Cbx7 tethering triggers a physiological sequence-independent feedback mechanism for PcG

protein recruitment in mESCs. We have included a brief discussion of this aspect in the revised version of our manuscript.

Reviewer #1: "... it is not clear that the findings provide any insight that is relevant for understanding how inheritance of Polycomb-dependent silencing works endogenously. One might be tempted to say that the authors have uncovered an important mechanistic difference in the respective functions of canonical PRC1 component Cbx7 and variant PRC1 component Rybp, but then again one might also be tempted to attribute that difference more trivially to the vastly different levels of expression of the TetR-FLAG fusion proteins as compared to their endogenous counterparts (Rebuttal Fig. 5a).

Response: We appreciate the reviewer's reservations about using artificial conditional tethering approaches to model the physiological mechanism of epigenetic inheritance. However, for an otherwise intractable question, these synthetic approaches present the current state-of-the-art having revealed fundamental insight to epigenetic feedback mechanisms in yeast and mammalian cells (Hathaway et al., Cell 2012; Rangunathan et al., 2015; Audergon et al., 2015).